# 2-Alkyl-Substituted-4-Amino-Thieno[2,3-*d*]Pyrimidines: Anti-Proliferative Properties to In Vitro Breast Cancer Models

**DOI:** 10.3390/molecules28176347

**Published:** 2023-08-30

**Authors:** Ivan Iliev, Anelia Mavrova, Denitsa Yancheva, Stefan Dimov, Galya Staneva, Alexandrina Nesheva, Iana Tsoneva, Biliana Nikolova

**Affiliations:** 1Institute of Experimental Morphology, Pathology and Anthropology with Museum, Bulgarian Academy of Sciences, Acad. G. Bonchev Str., bl. 25, 1113 Sofia, Bulgaria; taparsky@abv.bg; 2Department of Organic Chemistry, Faculty of Chemical Technologies, University of Chemical Technology and Metallurgy, S8 Kliment Ohridski Blvd., 1756 Sofia, Bulgaria; anmav@abv.bg (A.M.); s_t_e_v_e@abv.bg (S.D.); 3Institute of Organic Chemistry with Centre of Phytochemistry, Bulgarian Academy of Sciences, Acad. G. Bonchev Str., bl. 9, 1113 Sofia, Bulgaria; denitsa.pantaleeva@orgchm.bas.bg; 4Institute of Biophysics and Biomedical Engineering, Bulgarian Academy of Sciences, Acad. G. Bonchev Str., bl. 21, 1113 Sofia, Bulgaria; nescheva@gmail.com (A.N.); itsoneva@bio21.bas.bg (I.T.)

**Keywords:** 4-amino-thieno[2,3-*d*]pyrimidines, synthesis, breast cancer cell lines MCF-10A, MCF-7, MDA-MB-231, anti-proliferative activity, cell cycle analysis, zeta potential, pharmacokinetics descriptors

## Abstract

Thienopyrimidines are structural analogs of quinazolines, and the creation of new 2-alkyl derivatives of ethyl 4-aminothienopyrimidine-6-carboxylates for the study of their anti-proliferative properties is of great pharmacological interest. Some 2-alkyl-4-amino-thieno[2,3-*d*]pyrimidines **2**–**5** were synthesized, and their cyto- and phototoxicity against BALB 3T3 cells were established by an in vitro 3T3 NRU test. The obtained results indicate that the tested compounds are not cytotoxic or phototoxic, and that they are appropriate to be studied for their anti-proliferative and anti-tumor properties. The anti-proliferative potential of the compounds was investigated on MCF-7 and MDA-MB-231 cancer cells, as well as a MCF-10A cell line (normal human mammary epithelial cells). The most toxic to MCF-7 was thienopyrimidine **3** with IC_50_ 13.42 μg/mL (IC_50_ 0.045 μM), followed by compound **4** (IC_50_ 28.89 μg/mL or IC_50_ 0.11 μM). The thienopyrimidine **4** revealed higher selectivity to MCF-7 and lower activity (IC_50_ 367 μg/mL i.e., 1.4 μM) than compound **3** with MCF-10A cells. With respect to MDA-MB-231 cells, ester **2** manifested the highest effect with IC_50_ 52.56 μg/mL (IC_50_ 0.16 μM), and 2-ethyl derivative **4** revealed IC_50_ 62.86 μg/mL (IC_50_ 0.24 μM). It was estimated that the effect of the substances on the cell cycle progression was due to cell cycle arrest in the G2 stage for MDA-MB-231, while arrest in G1 was detected for the estrogen (ER)-positive MCF-7 cell line. The tested compound’s effects on the change of the zeta potential in the tumorigenic cells utilized in this study were determined. The calculation which we performed of the physicochemical properties and pharmacokinetic parameters influencing the biological activity suggested high intestinal absorption, as well as drug-likeness.

## 1. Introduction

Cancer is a disease caused by mutations in certain genes, and, according to the WHO, it is the second leading cause of death worldwide. The number of cancer-related deaths is expected to reach 11.5 million in 2030. Although significant advances have been made in the treatment of cancer, side effects and drug resistance remain unsolvable problems in the fight against cancer. Therefore, selective and effective chemotherapeutic agents for cancer treatment are still being sought [1,2,3].

Breast cancer, in particular, is the most prevalent cause of cancer death in women, accounting for 23% of all cancers and 14% of cancer deaths. Based on these concerning data, it should be noted that research in this area is important not only for the discovery of new drugs and methods of treatment, but also to overcome the psychological load. The isolation of MCF-7 cells (Michigan Cancer Fondation-7) from pleural effusion in the 1970s led to the formation of the MCF-7 cell line, commonly used in many studies [4,5]. The MDA-MB-231 (M.D. Anderson—Metastatic Breast 231) cell line is a highly aggressive epithelial human breast cancer cell line, which is also extracted from pleural effusion of a female with mammary adenocarcinoma. This cell line has served as a basis for many investigations dedicated to the creation of new anti-cancer agents [6]. Depending on the breast cancer type, variable therapy treatments can be used, such as immunotherapy, targeted drug therapy, and hormone receptor-positive breast cancer therapy. The list of the utilized clinic drugs was significantly expanded in the last decade. Nevertheless, active research to discover additional specific targets, as well as new chemotherapeutics, is going on [7,8,9].

The significance of the rational drug design is the basis of many studies aiming to generate novel, small, drug-like molecules. After the development of 4-aminoquinazolines lapatinib, gefitinib, and trastuzumab as anti-breast-cancer agents, a number of thienopyrimidines were synthesized as bioisosteres in order to study their anti-proliferative properties [10,11]. Structural combinations of chalcone with thieno[2,3-*d*]pyrimidinone scaffolds were generated, and their inhibitory activity against MCF-7 cells was proven. Some of the compounds showed anti-cancer and anti-proliferative activities at the G1 phase against MCF-7 cells, which were comparable to those of uracil [12]. 2,3-Disubstituted thieno[2,3-*d*]pyrimidin-4(3H)-ones prepared through one-pot synthesis were reported to inhibit the proliferation of A549 and MCF-7 cells [13]. A series of 2-trifluoromethylthieno[2,3-*d*]pyrimidine derivatives were designed and screened for their anti-tumor effects on MCF-7 and HepG2. Within the series, some derivatives exhibited higher activity than Gefitinib, a selective inhibitor of epidermal growth factor receptor’s (EGFR) tyrosine kinase [14,15].

Research in the field of ligand-based design led to the discovery of the thienopyrimidine derivatives apitolisib and CUDC-907 inhibiting PI3K kinase and mTOR kinase [16,17,18,19].

4-Hydroxy substituted thieno[2,3-*d*]pyrimidines containing isoxazole heterocycle were tested for their cytotoxicity to A549, HCT116 and MCF-7 cell lines. The assay indicated that 6-methyl-4-{[3-(4-chlorophenyl)-isoxazol-5-yl-]-methoxy}-thieno[2,3-*d*]pyrimidine exhibits a higher IC_50_ value (4.21 × 10^−3^ μM) in comparison to gefitinib (IC_50_ 20.68 μM) [20]. 5-Amino-2-ethylmercapto-4-phenyl-6-subistituted thieno[2,3-*d*]pyrimidine derivatives were designed, synthesized, and studied for their cytotoxicity, then screened against three human breast cancer lines. The most promising compound manifested strong cytotoxicity against the MCF-7 cell line, with IC_50_ values of 8.3 μg/mL [21]. The cytotoxic effects of some thienopyrimidines were tested against a human breast cancer cell line (MCF-7) and compared with that of doxorubicin [22].

It was established that 2-(benzylamino)-5,6-dimethylthieno[2,3-*d*]pyrimidin-4(3H)-one demonstrated cytotoxic activity on the breast cell line MDA-MB-435 (growth percent (GP) = −31.02%) [23]. New 4-substituted-aminothieno[2,3-*d*]pyrimidine derivatives were synthesized via the reaction of the corresponding 4-chlorothieno[2,3-*d*]pyrimidine derivatives with N-methylpiperazine, morpholine, N-phenylpiperazine, or 1,3-propanediamine. The 4-substituted aminothieno[2,3-*d*]pyrimidine scaffold is another promising backbone for the design and synthesis of potent cytotoxic leads [24]. It has been reported that some 5,6,7,8-tetrahydrobenzo[4,5]thieno[2,3-*d*]pyrimidine derivatives show significant potency against MDA-MB-231 and can avoid resistance mediated by Pgp and βIII-tubulin. N4-(4-methoxyphenyl)-N4-methyl-5,6,7,8-tetrahydrobenzo[4,5]thieno[2,3-*d*]pyrimidine-2,4-diamine has demonstrated significant anti-tumor effects in an in vivo xenograft model (MDA-MB-435) [25]. Many investigations on thienopyrimidine’s anti-cancer properties have been conducted and published [26,27,28,29].

Despite the results which have been achieved, no medicaments effective enough for the treatment of breast cancer have been developed yet. That is why the creation of new small molecules with good efficiency is of great importance. As can be seen from Figure 1, the thienopyrimidines can be considered as bioisosteres of the biological nitrogenous bases uracil and adenosine.

Based on the aforementioned properties of thienopyrimidines, we focused our study on the synthesis of some ethyl 2-alkyl-substituted-thienopyrimidine carboxylates through structure modification of derivatives A and B (Figure 2) in order to evaluate their effects against MCF-7 and MDA-MB-231 cancer cells. The choice of these structures is of pharmacological interest, because thienopyrimidines containing an amino group in the 4th and an alkyl fragment in the 2nd position (C) are less studied than the relevant thieno[2,3-*d*]pyrimidines A and B. Furthermore, these substituents occur in some target breast cancer inhibitors. The current study is a continuation of our previously reported investigations [30,31,32]. Herein, we also report the cytotoxicity and phototoxicity of the tested compounds against BAL3 cells, as well as the substance effects on the cell cycle progression of ER-negative MDA-MB-231 and ER-positive MCF-7 cell lines.

## 2. Results and Discussion

### 2.1. Chemistry

As mentioned above, the aim of the study was to generate hybrid molecules by introducing an amino group in the 4th position of the thieno[2,4-d]pyrimidine nucleus, as well as to induce ethoxycarbonyl and methyl groups in the thiophene ring instead of the fused tetrahydrobenzene fragment. The aryl substituents at the 2nd position were replaced with 2-ethoxy-2-oxoethylene, 2-chloroethyl, ethyl, and methyl groups. 2-Alkyl-substituted-4-amino-thieno[2,3-d]pyrimidines were synthesized as shown in Figure 1.

Ethyl 2-amino-3-cyano-4-methyl-thiophene-5-carbioxilate **1** was synthesized according to the Gewald reaction by means of ethyl acetoacetate, malononitrile, and sulfur in ethanol medium. Diethyl amine was added drop-wise as a catalyst by stirring at an ambient temperature [33,34,35]. The obtained tetra-substituted thiophene served as a precursor for cyclocondensation with different nitriles. The process leading to the target 4-amino-thieno[2,3-d]pyrimidines was conducted under acidic conditions, accomplished by passing dry hydrogen chloride gas through the reaction mixture. The structures of thienopyrimidines **2**–**5** were identified through IR, ^1^H-NMR spectra, and elemental analysis. The stretching vibration of the ester groups in the thiophene ring gave rise to IR bands within the range of 1722–1711 cm^−1^. In the ^1^H NMR spectra, the signals of the thiophen ester group -OCH_2_CH_3_ appeared in the form of a triplet at 1.3–1.4 ppm for the -CH_3_ group and a quartet at 4.0–4.4 ppm for the -CH_2_- group. The signals of the -CH_3_ group attached to the thiophen ring were detected as singlets at 2.85–2.87 ppm. In the ^13^C NMR spectra, the thiophen ester group -OCH_2_CH_3_ was characterized by signals at 61–62 ppm (for the -CH_2_-) and ca. 14 ppm (for the -CH_3_), while the signals for the methyl group at 5-position were observed at ca. 15 ppm. The differentiation of the methylene and methyl signals was clearly visible from the DEPT (distortions enhancement by polarization transfer) NMR spectra (Appendix A). The different combinations of methyl/methylene groups of the substituents in the pyrimidine rings also had very characteristic patterns: the methylene groups resonated within the range of 32–42 ppm, and the methyl groups of compounds **4** and **5** within a broader range, i.e., 12–22 ppm, respectively. The assignment of the H and C signals was confirmed by heteronuclear single quantum correlation (HSQC) spectra (Appendix A). The assignment of the NMR signals was also supported by the prediction of NMR isotopic shielding by B3LYP/6-311++G** calculations in DMSO solvent (GIAO method) for all compounds (Appendix A). The comparison between the calculated and experimental data demonstrated that C5 in the thiophen ring (bearing the methyl group) was characterized by a signal at ca. 140 ppm; C6 and βC by signals in the interval of 114–120 ppm; and C2, C4, αC, and the C-atom from the carbonyl group by signals in the interval of 160–169 ppm. The signals for the amino group protons were detected at ca. 7.3–7.4 ppm only in the spectra of compounds **2** and **4**, while in the spectra of **3** and **4,** they were lacking, most probably due to an exchange of protons with the water present in the solvent. Nevertheless, the presence of the amino group in all four compounds was unambiguously confirmed by the two IR absorption bands for N-H stretching vibrations in the ranges of 3490–3400 cm^−1^ and 3375–3280 cm^−1^, as well as the band for scissoring vibration of the amino group at ca. 1640–1660 cm^−1^, which was found in the IR spectra.

### 2.2. Biological Study

#### 2.2.1. In Vitro Safety Test

Compounds **2**–**5** were studied for cytotoxicity/phototoxicity by an in vitro 3T3 NRU-test. The cell line BALB 3T3, clone A31 (mouse embryo fibroblasts), was incubated with the test compounds at a concentration from 15 to 4000 µg/mL for 24 h at 37 °C, 5% CO_2_, and 95% humidity. The cytotoxicity and phototoxicity were expressed in percentages relative to the negative control determined. Dose–response dependence was observed for all of the tested thienopyrimidines. The obtained results are shown in Figure 3. Based on the dose–response curves, the CC_50_/PC_50_ values (50% cytotoxic/phototoxic concentration) were calculated by nonlinear regression analysis (Table 1). The CC_50_ values can be used to calculate the PIF (photo-irritancy factor) for each test compound according to the following formula: PIF = CC_50_/PC_50_. The PIF demonstrates the probability that the test compound may cause a phototoxic effect (PIF < 2: not phototoxic; PIF ≥ 2 < 5: probable phototoxicity; PIF ≥ 5: phototoxic). The PIF < 2 obtained for all tested compounds shows a high level of photo safety.

In our study, as a positive control for phototoxicity, acridine orange (AO) was used.

These results show that compounds **2**–**5** are safe for use in pharmaceuticals.

BALB 3T3 cells have the ability to divide indefinitely, but they are extremely sensitive to inhibition of cell division after fusion and highly susceptible to transformation. Therefore, they are widely used in carcinogenesis studies.

#### 2.2.2. Anti-Proliferative Effects

As can be seen from the data in Table 1, compounds **2**–**5** revealed neither cytotoxicity nor phototoxicity; thus, they were suitable to be screened for their anti-proliferative and anti-tumor properties using an MTT-test. The MCF-10A cell line (normal human mammary epithelial cells) served as a reference control.

All studied compounds showed lower anti-proliferative activity against MCF-10A in comparison to the manifested cytotoxicity of MCF-7 and MDA-MB-231, see Table 2 below. As the selectivity of anti-cancer agents plays a significant role in the treatment of cancer diseases, the selectivity index (SI) of the compounds was calculated and results are shown in Table 3. The MTT test results showed that the most toxic to MCF-7 was thienopyrimidine **3**, with an IC_50_ of 13.42 μg/mL (IC_50_ 0.045 μM, respectively), followed by compound **4** (IC_50_ 28.89 μg/mL or IC_50_ 0.11 μM). At the same time, with regard to MCF-10A, compound **4** demonstrated a cytotoxicity level eight times lower than compound **3**. Moreover, the ethyl substituted thienopyrimidine **4** had a selectivity index (SI = 12.7) greater than that of the 2-chloroethyl-substituted compound **3**, confirming compound **4** as a lead structure. Compound **5** demonstrated similar properties, which resulted in the highest SI. The calculated selective index in the positive control (Cisplatin) was SI = 25.

With respect to MDA-MB-231 cells, ester **2** manifested the highest anti-tumor activity, with an IC_50_ of 52.56 μg/mL (IC_50_ 0.16 μM), while the cytotoxicity of this compound to BALB3T3 was IC_50_ 1736 μg/mL (IC_50_ 5.37 μM). 2-Ethyl-4-amino-thieno[2,3-d]pyrimidine **4** exhibited a slightly lower anti-tumor effect, with an IC_50_ of 62.86 μg/mL (IC_50_ 0.24 μM) and cytotoxicity to BALB 3T3 of IC_50_ 2628 μg/mL (IC_50_ 9.37 μM).

The cytotoxicity levels of compounds **2**–**5** against MCF-10A, MCF-7, and MDA-MB-231 cells are presented graphically in Figure 4.

#### 2.2.3. Cell Cycle Arrest

Our study continued with the examination of the effect of the newly synthesized compounds’ IC_50_ values on cell cycle progression. The cell aliquots were collected, and FACS analyses were used to detect the effects of cell treatments with the selected compounds (**2**, **4**, and **5**). A propidium iodide staining assay was performed. The results are presented in Figure 5.

The changes in the cell cycle distribution in % (G1, S, and G2 phases), based on the graphical figures A–C, were calculated and are given in Table 4.

As shown in Table 4, 40.41% of the untreated MCF-7 cells were in the G1 phase of the cell cycle. The G1 phase occurred between the M and S phases, i.e., the stage in which cells are preparing to divide. After treatments, increases of 11%, 20%, and 20.5%, respectively, compared to the control were detected in this phase. Regarding the highly metastatic breast cancer cell line MDA-MB-231, 4.38% of the cells of the untreated control were in phase G2, a period of rapid cell growth and protein synthesis during which the cell prepares itself for mitosis. Dramatic increases in the percentage of cells in this phase were observed after the treatments, i.e., 53.65%, 44.9%, and 45.9%, which suggested the existence of a block at the G2/M phase of the cell cycle. The data presented in Table 4 demonstrate that the tested compounds have different effects on estrogen (ER)-positive and -negative breast cancer lines. The inhibitory effect on cell proliferation occurred due to cell cycle arrest in the G2 stage for the ER-negative MDA-MB-231, while arrest in G1 was detected for the ER-positive MCF-7 cell line.

#### 2.2.4. ζ-Potential Measurement

Surface electrical charge is specific for each cell type, and is especially important in malignant cells because it determines their state of aggregation and adhesion in the organs. Nearly all metabolically active cancer cells are known to secrete a large amount of lactate as mobile anions. The increased levels of glucose uptake and lactate secretion could be up to 30 times those of normal cells, which is the basis for cancer-detection technologies such as PET (positron emission tomography) scan diagnostics. Cancer cells may have slightly elevated surface contents of negatively-charged immobilized molecules (e.g., sialic acid), i.e., 30–50% more than normal cells [33,34,35,36,37,38,39].

The treatment of tumorigenic (MCF-7, MDA-MB-231) and non-tumorigenic (MCF-10A) cell lines with compounds **2**, **4**, and **5** demonstrated that these compounds were able to affect their ζ-potential, as shown in Figure 6. First, the ζ-potential of each cell line was measured, and all obtained values were negative (Figure 6A). The Student *t*-test statistical analysis which was applied distinguished the ζ-potentials of the three studied cell lines with *p* < 0.05, and the values followed the order of MCF-7 > MDA-MB-231 > MCF-10A. The cells treated with compounds **2**, **4**, and **5** exhibited more negative ζ-potential values. Figure 5B represents the relative percentage change of the ζ-potential induced by the studied compounds for the different cell lines. Compound **2** was able to decrease the ζ-potential of the cells with the following ranking: MCF-7 (42%) > MCF-10A (24%) > MDA-MB-231 (16%). For compound **4**, the ranking was as follows: MCF-10A (50%) > MDA-MB-231 (46%) > MCF-7 (3%). For compound **5**, this was MCF-10A (36%) > MCF-7 (22%) > MDA-MB-231 (23%).

The ζ-potential of all studied cell lines was negative: MCF-7 > MDA-MB-231 > MCF-10A (Figure 6), which is consistent with the data obtained for anti-proliferative activity (Table 3, Table 4 and Table 5). The investigated compounds **2**, **3**, and **5** showed lower anti-proliferative activity against MCF-10A in comparison to the manifested cytotoxicity against the tumorigenic MCF-7 and MDA-MB-231 cell lines. As mentioned above, cancer cells have a higher zeta potential due to the negatively charged mobile lactate products in elevated glycolysis. After treatment with compounds **2**, **4**, and **5**, the changes in the zeta potential were not of the same rank. MCF-10A showed higher electrical potential compared to the metastatic cells. One reason for this surprising effect could be the higher percentage of mitotic cells, as can be seen in Table 4, where the surface area is nearly double. The dielectric properties of a cell are determined by the distribution of surface charges, cell size, and morphology, as well as the conductivity and permittivity of their membranes, and could also affect the zeta potential. Henslee et al. have shown that the average radii (µm) are MCF-10A 9.25, MCF-7 9.1, and MDA-MB-231 8.93 with membrane capacitance (F/m^2^) of MCF-10A 0.0194, MCF-7 0.0186, and MDA-MB-231 0.0163. This describes the specificity of the dipolophoretic electric properties of breast cancer models [4,39]. On the other hand, cell membranes’ electrical charges are also affected by the sialic acid present in glycolipids and glycoproteins [38,40], as well as the different number of microvili on the cell surface [39,41]. Therefore, it could be speculated that the changes occurring after treatment with the three thienopyrimidines are related to the altered sialic acid, the synthesis of more microvilli rich in lipid rafts, and the externalization of a higher content of negatively charged phosphatidyl serine from the inner to the outer part of the bilayer membrane. The fact that surface electrical properties also depend on phospholipid metabolism should not be overlooked. The membrane of MCF-10A is more ordered in comparison to the tumorigenic MCF-7 and MDA-MB-231, which has been confirmed by reports in the literature that the membranes of malignant cells are more fluid compared to the membranes of non-cancerous cells. The highly metastatic cell line MDA-MB-231, however, showed a higher order compared to MCF-7, possibly due to the higher cholesterol content of the highly metastatic cell line [40,42,43].

As a conclusion, regarding the changes in the zeta potential when treated with 2-alkyl-substituted-4-amino-thieno[2,3-d]pyrimidines **2**, **4**, and **5**, it can be suggested that the observed changes are not only related to the effects caused by the anti-proliferative activity of the aforementioned compounds. It could be further assumed that the specific physico-chemical features of the three types of cells with different metastatic potential, such as different dielectrophoretic potential, phospholipid composition, microvilli, or cell cycle distribution, play an additional role in the surface electrical properties of cells after compound treatment (Figure 6).

### 2.3. Physico-Chemical Properties and Drug-Likeness Analysis

An important goal of drug research is to gain sufficient understanding of the molecular properties that limit oral bioavailability in order to facilitate the design of viable new drug candidates. Lipophilicity and the number of hydrogen bond donors appear to be key properties, as they remain essentially constant in oral medications over time. The number of H-donors and H-acceptors affects the polar surface of the molecule and reduces the rate of its membrane penetration, which correlates with the bioavailability. Therefore, it is important to calculate the lipophilicity and the physico-chemical parameters of the studied compounds which affect the absorption, distribution, metabolism, excretion, and toxicological properties. The physico-chemical properties of the compounds given in Table 5 were generated using the Molinspiration software, v2013.09 [41,44]. The descriptors of the synthesized ligands (Table 5) are consistent with Lipinski’s rule of five, without any violations [42,45]. Equally important to the characterization of a ligand as a drug candidate is the determination of ligand efficiency (LE). LE is related to the Gibbs free energy of binding for a heavy atom, and may serve to establish the preferred pharmacophores of the leads. The calculation of LE can be performed using the equation given below (1) [43,44,45,46,47,48,49]:

LE = pIC_50_ × 1.37/HA (kcal/mol)
(1)


Ligand lipophilicity efficiency (LLE), also known as lipophilic efficacy (LipE), is related to the possibility of transferring the ligand from 1-octanol to the active site of binding. It has been proposed as a better alternative to LE to capture the enthalpic component of ligand binding [47,48,50,51]:

LLE = pIC_50_ − cLogP
(2)


Since lipophilicity and effectiveness play important roles in the development of new drug candidates, it is essential that their values be greater than 0.3 and 3, respectively [49,52]. As can be seen, the LE and LLE values of compounds **2**–**5,** given in Table 5, are higher than 0.3 and 3.

The absorption of a drug intended for oral intake depends on its dissolution, i.e., on the solubility of the chemical compound. For this reason, solubility is another important parameter that should be considered when describing biological properties and bioavailability, in particular. SwissADME [50,53] was used to calculate the possible solubility [51,52,53,54,55,56] and to determine the drug-likeness of the studied compounds. The obtained results regarding the solubility of compounds **2**–**5** are given in Table 6.

Log S (ESOL−Estimated SOLubility) is a method for the estimating the aqueous solubility of a compound directly from its structure [51,54]. Log S (Ali) solubility is calculated by an alternative model for predicting aqueous solubility by incorporating TPSA [52,55]. Both methods differ from the basic solubility equation, avoiding the melting point parameter.

The third method for the establishment of solubility was developed by SILICOS-IT. The linear correlation coefficient was corrected by molecular weight, and its value was R2  =  0.75 [54,57]. The results obtained by the use of the Ali method revealed that all compounds exhibited moderate solubility. The calculations by the other two methods determined the compounds as soluble, except for compound **3**. These results are a sign that the compounds can be considered as orally applicable candidates.

The pharmacokinetic properties of a drug include the relationship between its absorption, distribution, and inactivation. The passage of drugs across cell membranes is a key part of most pharmacokinetic processes. The most important route by which a drug crosses cell membranes is passive diffusion, the rate of which is determined by molecular size, concentration gradient, lipid solubility, degree of drug ionization, and protein binding. The calculated results, given in Table 7, indicate that thienopyrimidines **2**–**5** could exhibit high gastrointestinal absorption.

According to the calculation, a high gastrointestinal absorption (GI) was characteristic of all compounds, although they may not pass the blood–brain barrier (BBB). On the other hand, the compounds were predicted not to act as substrates of Pg (protein that pumps substances out of cells), which is beneficial, as if they were to enter the central nervous system, they would not be eliminated by the Pg protein. The studied thienopyrimidines might behave as CYP (cytochrome P450 superfamily) substrates/inhibitors towards 1A2, 2C19, and 2C9, but not to members of the 2C6 and 2C4 CYP families.

The data related to drug-likeness correspond to five rules, bearing the names of their authors. For Ghose et al. [54,57], the calculated logP should be between −0.4 and 5.6, with an average value of 2.52. For molecular weight, the qualifying range is between 160 and 480, with an average value of 357. For molar refractivity, the qualifying range is between 40 and 130, with an average value of 97. According to Egan’s rule, a compound demonstrates good bioavailability when its TPSA value is within the range of 0 ≥ tPSA ≤ 132 Å2 and −1 ≥ logP ≤ 6 [55,58]. In researching the molecular properties affecting oral bioavailability, Veber et al. concluded that the polar surface area (PSA) should be ≤140 Å2, while a number of rotatable bonds ≤ 10 or a sum of H-bond donors and acceptors ≤ 12 are sufficient and selective criteria for determining a drug candidate [56,59]. The results in Table 5 show that the physico-chemical characteristics of the compounds were in accordance with the aforementioned rules. Muegge’s rule is based on the so-called pharmacophore point filter, the role of which is to distinguish substances that are drug-like and those that are not, which is based on simple structural rules. To pass filtering, a candidate drug must receive two to seven pharmacophore points [57,60]. For compounds with a carboxylic acid, amine, amidine, or guanidine pharmacophore point, the survival threshold is lowered to one.

The compounds which pass the PAINS (pan-assay interference compounds) filter or pan-assay interference compounds are prominent sources of false positives in the drug-discovery process [61].

The bioavailability radar in Figure 7 is a graphical depiction of the drug-likenesses of compounds **2**–**5**.

As can be seen from Figure 7, the values of the lipophilicity, size, polarity, solubility, saturation, and flexibility of compounds **2**–**5,** summarized in the SwissADME radar of bioavailability, support that the compounds meet the requirements for drug likeness.

Bioactive molecules exert their action by interacting with specific target proteins or other macromolecules. The prediction of these targets restricts their numbers, thus contributing to the faster and more relevant study of the biological properties of a molecule [59,62]. It would be expected that thienopyrimidine 4 would reveal its action by interacting with kinase enzymes (Figure 8). A kinase is a type of enzyme that is involved in a number of cellular processes. Some kinases are thought to cause cancer.

The obtained results regarding the anti-proliferative activity of compound **4** against MCF-7 and MDA-MB-231 showed lower cytotoxicity compared to the most active compound of series B (Ar = Phen, Figure 2), but thienopyrimidine **4** had lower cytotoxicity compared to BALB 3T3. On the other hand, this compound showed a higher selective action index towards the investigated cancer cells. By performing SwissAdme calculations, it was found that the parent compound exhibited poor solubility and a XLOGP3 > 3.5, which was a violation of the criterion for lead-likeness.

## 3. Materials and Methods

For the synthesis of the target compounds, the following commercial products were used without further purification: ethyl acetoacetate, chloropropionitrile (Merck, Rahway, NJ, USA), ethyl cyanoacetate, acetonitrile, propanonitrile, and malononitrile (Alpha Aesar, Karlsruhe, Germany). All inorganic substances and organic solvents utilized were pure for synthesis or analysis (Macron, Merck).

Melting points were determined as the phase transition, from solid to liquid, at atmospheric pressure on a Boetius PHMK 5 microscope heating table, in degrees Celsius ±1.0 °C. The purity levels of all the obtained compounds, as well as the retention coefficients (Rf), were estimated on F254 or Al_2_O_3_ 60 silica gel plates (Merck, 0.2 mm). IR spectra were measured using KBr tablets on a Varian Scimitar 1000 spectrophotometer, or by the ATR technique on a Bruker Equinox 55 spectrophotometer. All ^1^H-NMR spectra were recorded in DMSO-d6 solvent on a Bruker NEO 400 spectrometer (Bruker, Faelanden, Switzerland). Chemical shifts were expressed to tetramethylsilane (TMS) and were represented in δ (ppm).

### 3.1. Synthesis

#### 3.1.1. General Procedure for the Synthesis of Ethyl 2-Amino-3-Cyano-4-Methylthiophene-5-Carboxylate **1**

To a solution of 5.5 mL (6.6 g, 0.1 mol) malonodinitrile and 12.7 mL (13.01 g, 0.1 mol) ethyl acetoacetate in 30 mL ethanol, 3.2 g sulfur was added. Via cooling and vigorous stirring, an equimolar amount of the catalyst (diethylamine, 0.1 mol) was dripped for 30 min. The reaction mixture was stirred for a further 75 min, the solution was cooled, and the amino-thiophene was crystallized. The precipitate was filtered off, washed thoroughly with water, recrystallized from ethanol, and dried in a vacuum. Yield: 53%; Mp—209–210 °C; Rf—0.67 (Benzene/MeOH = 5:1; IR (KBr, cm^−1^): 3401 (νNH), 3313 (νNH), 3202 (νNH), 2981 (νCH3), 2932 (νCH2), 2904 (νCH3), 2204 (νCN), 1674 (νC=O), 1631 (νArH), 1546 (νArH), 1494 (νArH), 1416 (δCH3); 1313 (C-N), 1272 (C-N), 1191, 1107 (νC-O).

#### 3.1.2. General Procedure for the Synthesis of 2-Substituted 4-Aminothieno[2,3-d]Pyrimidines **2**–**5**

To a solution of 0.0094 mol of 2-amino-thiophene-3-carbonitrile in 20 mL dry dioxane, 0.0094 mol of the corresponding RCN derivative was added. Dry hydrochloric gas was passed through the reaction mixture for 6 h via continuous stirring at room temperature. The solution was allowed to stand at room temperature for 12 h, then poured onto ice and neutralized with 10% (*v*/*v*) NH_4_OH to pH~8. The precipitate which was formed was filtered off, washed extensively with water, and dried in a vacuum dryer at 60 °C.

##### Ethyl 4-Amino-2-(2-Ethoxy-2-Oxoethyl)-5-Methylthieno[2,3-d]Pyrimidine-6-Carboxylate (**2**)

During neutralization, a resinous sludge was observed at pH~6. The non-resinous portion was transferred to another flask and neutralized. The neutralization product was purified by recrystallization with methanol. The resinous portion was allowed to stay in 25% (*v*/*v*) NH_4_OH for 12 h in order to crystallize. The obtained crystals were recrystallized with benzene. White crystals were obtained. Yield: 28%; Mp. 206–207 °C; Rf—0.70 (Benzene/MeOH = 5:1) IR (KBr, cm^−1^): 3452 (νNH), 3323 (νNH), 3140 (νNH), 2978 (νCH), 2935 (νCH), 1714 (νC=O), 1660 (δNH), 1245 (νC–O); ^1^H NMR (DMSO-d6, δ ppm): 1.16–1.20 (t, *J* = 6.2 Hz, ^3^H, C_2_-CH_2_-COOCH_2_CH_3_); 1.28–1.32 (t, *J* = 6.1 Hz, ^3^H, C_6_-COOCH_2_CH_3_); 2.86 (s, ^3^H, CH_3_-C_5_); 3.72 (s, ^2^H, -C_2_-CH_2_-C=O); 4.07–4.12 (q, *J* = 6.4 Hz, ^2^H, C_6_-COOCH_2_CH_3_); 4.28–4.32 (q, *J* = 6.6 Hz, ^2^H, C_2_-CH_2_-COOCH_2_CH_3_); 7.4 (bs, ^2^H, NH_2_); ^13^C NMR (DMSO-d6, δ ppm): 14.51 (C_6_-COOCH_2_CH_3_); 14.58 (C_2_-CH_2_-COOCH_2_CH_3_); 15.92 (CH_3_-C_4_); 45.16 (-C_2_-CH_2_-C=O); 60.88 (C_6_-COOCH_2_CH_3_); 61.60 (C_2_-CH_2_-COOCH_2_CH_3_); 114.98 (βC); 120.33 (O=C-C_6_); 140.74 (CH_3_-C_5_); 160.60 (CH_2_-C_2_); 162.24 (αC); 162.73 (H_2_N-C_4_); 168.0 (C_2_-CH_2_-C=O); 169.85 (-CH_2_OC=O-C_6_); calculated for: C_14_H_17_N_3_O_4_S; C, 52.00; H, 5.30; N, 12.99; O, 19.79; S, 9.92; found: C, 52.08; H, 5.38; N, 13.02; O, 19.76; S, 9.96.

##### Ethyl 4-Amino-2-(2-Chloroethyl)-5-Methylthieno[2,3-d] Pyrimidine-6-Carboxylate (**3**)

The substance was purified by recrystallization from EtOAc. Yield: 59%; Mp. 163–165 °C; Rf = 0.76 (EtOAC/n Hexane = 5:1); IR (KBr, cm^−1^): 3419 (νNH), 3278 (νNH), 3173 (νNH), 2988 (νCH3), 2840 (νCH2), 1722 (νC=O), 1652 (δNH), 1251 (νC–O); ^1^H NMR (DMSO-d6, δ ppm): 1.29–1.33 (t, *J* = 7.1 Hz, ^3^H, -OCH_2_CH_3_); 2.87 (s, ^3^H, CH_3_-C_4_); 3.23–3.26 (t, *J* = 6.5 Hz, ^2^H, -CH_2_-C_2_); 4.07–4.10 (t, *J* = 6.6 Hz, ^2^H, -CH_2_-Cl); 4.9–4.34 (q, *J* = 7.1 Hz, ^2^H, -OCH_2_CH_3_); ^13^C NMR (DMSO-d6, δ ppm): 14.57 (-OCH_2_CH_3_); 15.72 (CH_3_-C_5_); 39.76 (CH_2_-C_2_); 42.31 (CH_2_Cl), 61.90 (-OCH_2_CH_3_); 104.73 (βC); 115.41(O=C-C_6_); 121.79 (CH_3_-C_5_); 140.82 (αC); 159.17 (CH_2_-C_2_); 161.80 (NH_2_-C_4_); 162.28 (-O-C=O); calculated for C_12_H_14_ClN_3_O_2_S: C, 48.08; H, 4.71; Cl, 11.83; N, 14.02; O, 10.67; S, 10.70; found: C, 49.00; H, 4.69; Cl, 11.85; N, 14.06; O, 10.62; S, 10.74.

##### Ethyl 4-Amino-2-Ethyl-5-Methylthieno[2,3-d]Pyrimidine-6-Carboxylate (**4**)

In the process of neutralization, a part of a resinous product was observed. The liquid layer was transferred to another flask and neutralized. The product was obtained in the form of white crystals, and was isolated by recrystallization with MeOH. Yield: 36%; Mp. 173–175 °C; Rf—0.50 (EtOAc/n-hexane = 1:1); IR (KBr, cm^−1^): 3490 (νNH), 3304 (νNH), 3138 (νNH), 2969 (νCH3), 2937 (νCH2), 1711 (νC=O), 1639 (δNH), 1233 (νC–O); ^1^H NMR (DMSO-d6, δ ppm): 1.19–1.25 (t, *J* = 7.5 Hz, ^3^H, -CH_2_CH_3_); 1.28–1.32 (t, *J* = 7.0 Hz, ^3^H, -OCH_2_CH_3_); 2.62–2.70 (q, *J* = 7.6 Hz, ^2^H, -CH_2_CH_3_); 2.84 (s, ^3^H, CH_3_-C_5_); 4.26–4.32 (q, *J* = 7.0 Hz, ^2^H, -OCH_2_CH_3_); ^13^C NMR (DMSO-d6, δ ppm): 12.84 (C_2_-CH_2_CH_3_); 14.62 (-OCH_2_CH_3_); 15.97 (CH_3_-C_5_); 32.08 (-CH_2_CH_3_); 61.46 (-OCH_2_CH_3_); 114.65 (βC); 119.53 (O=C-C_6_); 140.82 (H_3_C-C_5_); 160.53 (αC); 162.84 (CH_2_-C_2_); 168.36 (NH_2_-C_4_); 169.60 (-O-C=O); calculated for: C_12_H_15_N_3_O_2_S; C, 54.32; H, 5.70; N, 15.84; O, 12.06; S, 12.08; found: C, 54.29; H, 5.74; N, 15.86; O, 12.03; S, 12.10.

##### Ethyl 4-Amino-2,5-Dimethyl-Thieno[2,3-d]Pyrimidine-6-Carboxylate (**5**)

A dry HCl gas was passed through a solution of 2-amino-thiophene-3-carbonitrile **1** (0.0094 mol) in pre-dried acetonitrile (30 mL) by stirring at room temperature for 5 h. The white precipitate which formed was filtered off, washed with water, dried, and recrystallized from methanol. Yield: 73%;Mp. 203–204 °C; Rf—0.66 (Benzene/MeOH = 5:1); IR (KBr, cm^−1^): 3375 (νNH), 3242 (νNH), 3169 (νNH), 2983 (νCH_3_), 2937 (νCH_2_), 2503 (ν=NH^+^), 1717 (νC=O), 1650 (δNH), 1593 (νArH), 1245 (νC–O); ^1^H NMR (DMSO-d6, δ ppm) 1.30–1.32 (t, *J* = 7.1 Hz, ^3^H, -OCH_2_CH_3_); 2.56 (s, ^3^H, CH_3_-C_2_); 2.87 (s, ^3^H, CH_3_-C_5_); 4.30–4.35 (q, *J* = 7.1 Hz, ^2^H, -OCH_2_CH_3_); ^13^C NMR (DMSO-d6, δ ppm): 14.53 (-OCH_2_CH_3_); 15.54 (CH_3_-C_5_); 22.40 (CH_3_-C_2_); 62.12 (-OCH_2_CH_3_); 115.20 (βC); 122.64 (O=C-C_6_); 140.82 (CH_3_-C_5_); 159.50 (αC); CH_3_-C_2_); 161.94 (NH_2_-C_4_); 161.94 (O-C=O); calculated for C_11_H_13_N_3_O_2_S: C, 52.57; H, 5.21; N, 16.72; O, 12.73; S, 12.76; found: C, 52.60; H, 5.23; N, 16.69; O, 12.71; S, 12.78.

### 3.2. Biological Assay

#### 3.2.1. Chemicals

The reference chemical (phototoxic/non-phototoxic drugs), acridine orange (Loba chemie Ltd., Fischamend, Austria), was purchased from Sigma-Aldrich, Schnelldorf, Germany. The cell culture reagents were Dulbecco’s modified Eagle’s medium (DMEM), fetal bovine serum (FBS), and antibiotics (penicillin and streptomycin). The disposable consumables were supplied by Orange Scientific, Braine-l’Alleud, Belgium.

#### 3.2.2. Cell Lines

All breast cell lines (MCF-10A, MCF-7, and MDA-MB-231) were purchased from the American Type Culture Collection—ATCC (Manassas, VA, USA). The MCF-7 and MDA-MB-231 were cultivated in Dulbecco’s modified Eagle’s medium (DMEM) and supplemented with 10% fetal bovine serum (FBS), 1% sodium pyruvate, and 1% MEM non-essential amino acids (NEAA) without antibiotics. The non-tumorigenic breast cell line (MCF-10A) was cultivated in DMEM medium (Sigma-Aldrich, St. Louis, MO, USA) and supplemented with 10% FBS, 1% sodium pyruvate, 1% non-essential amino acids (NEAA), 20 ng/mL human epithelia growth factor (hEGF), 10 μg/mL insulin, and 0.5 μg/mL hydrocortisone without antibiotics. All cell lines were maintained in a humidified atmosphere, with 5% CO_2_ at 37 °C.

#### 3.2.3. Cytotoxicity and Phototoxicity Testing

BALB/3T3 cells were cultured in 75 cm^2^ tissue culture flasks in DMEM, 10% FBS, 2 mM glutamine, and antibiotics (penicillin 100 U/mL and streptomycin-100 µg/mL) at 37 °C, 5% CO_2_, and 90% relative humidity. Cytotoxicity/phototoxicity was assessed by a validated BALB/3T3 clone A31 neutral red uptake assay (3T3 NRU test) [60,63]. Briefly, cells were plated in a 96-well microtiter plate at a density of 1 × 10^4^ cells/100 µL/well and incubated for 24 h. A wide concentration range was applied for the test compounds. For the phototoxicity tests, 96-well plates were irradiated with a dose of 2.4 J/cm^2^, and the cells were incubated for an additional 24 h. After treatment with neutral red medium, washing, and treatment with the ethanol/acetic acid solution, the absorption was measured on a TECAN microplate reader (TECAN, Grödig, Austria) at a wavelength of 540 nm. The light source which we used was a light-emitting diode (LED) matrix—an artificial solar light simulator Helios-iO, model LE-9ND55-H—5500K (SERIC Ltd., Tokyo, Japan).

Cytotoxicity/phototoxicity were expressed as CC_50_/PC_50_ values (concentrations required for 50% cytotoxicity/phototoxicity), calculated using non-linear regression analysis (GraphPad Software GraphPad Prism 9.5.1, San Diego, CA, USA). The CC_50_ values were used to calculate the PIF (photo-irritancy factor) for each test substance, according to the following formula:

PIF = Cytotoxicity (CC_50_)/Phototoxicity (PC_50_)

The statistical analysis included the application of one-way ANOVA, followed by a Bonferroni’s post hoc test, and *p* < 0.05 was accepted as the lowest level of statistical significance. All results are presented as mean ± SD.

#### 3.2.4. In Vitro Anti-Proliferative Activity

Using the standard MTT-dye reduction assay described by Mosmann [62,64], the anti-proliferative activity was tested. The method is based on the metabolism of the tetrazolium salt MTT to insoluble formazan. The formazan absorption was registered using a microplate reader at λ = 540 nm. The measured absorption is an indicator of cell viability and metabolic activity. Anti-proliferative activities were expressed as IC_50_ values (concentrations required for 50% inhibition of cell growth), calculated using non-linear regression analysis (GraphPad Software, San Diego, CA, USA).

#### 3.2.5. Cell Cycle Analysis by Propidium Iodide (PI) Staining

The cells were trypsinized, suspended in medium + 10% FCS, and centrifuged (1000 rpm, 5 min); then, the pellet was suspended in PBS (1 mL) and fixed with EtOH. The cells were suspended in 2.5 mL pre-cold 70% EtOH and incubated on ice for 15 min. The staining procedure was as follows: the pellet was suspended in a 500 μL PI-solution in PBS and 0.1 mg/mL RNase A, and 0.05% Triton X-100 was added. After incubation for 10 min at 37 °C, the cells were transferred to the flow cytometer and cell fluorescence was measured. The samples were measured by flow cytometry (LSR II, BD Biosciences, Franklin Lakes, NJ, USA) and analyzed using BD FACS DIVA software v. 6.1.3 (BD Biosciences).

#### 3.2.6. ζ-Potential Measurement

The zeta (ζ ξ) potential is the electrostatic potential at the shear plane of a cell, and is related to both its surface charge and the local environment. The ζ-potential of the intact cells and the treated ones was recorded in a cell suspension (0.5 × 10^6^ cells/mL) in PBS, using the electrophoretic light-scattering technique on a Zetasizer Nano ZS analyzer (Malvern Instruments, Great Britain). Laser Doppler electrophoresis and velocimetry of cells were used to determine the ζ-potential by measuring the speeds of the cells in PBS upon the application of an electric field. The ζ-potential was calculated based on the Helmholtz–Smoluchowski equation. The measurements were performed in a U-shaped cell with gold-plated electrodes at 37 °C.

The statistical analysis included the application of one-way ANOVA followed by a Bonferroni’s post hoc test, and *p* < 0.05 was accepted as the lowest level of statistical significance. All results are presented as mean ± SD. All experiments were performed in triplicate.

## 4. Conclusions

New 2-alkyl-subsrituted 4-amino-thieno[2,3-d]pyrimidines were synthesized and evaluated for their cytotoxicity and anti-proliferative effects. The results of the cytotoxicity and phototoxicity test which we performed showed that the studied compounds were not toxic towards BALB 3T3 cells. Compounds **3** and **4** demonstrated the highest anti-proliferative activity towards MCF-7; the IC_50_ values were 0.045 µM for 3 and 0.11 μM for **4**, respectively, but the thienopyrimidine **4** showed a much higher selectivity index (SI-12.7 vs. SI-3.38) and lower cytotoxicity against MCF-10A cells, which makes this compound preferable for further research. The highest anti-proliferative effect against MDA-MB-231 was demonstrated by compound **2**, with IC_50_ 0.16 μM and SI 9.6, followed by the thienopyrimidine **4**, with IC_50_ IC_50_ 0.24 μM and SI 5.8. On the basis of the cell cycle analysis, it was established that the revealed inhibitory effects of thienopyrimidines **2**, **4,** and **5** on cell proliferation of the tested cells was due to cell cycle arrest in the G2 stage for the ER negative MDA-MB-231, while for the ER positive MCF-7 cell line, the arrest was detected in the G1 phase. The results obtained by the treatment of tumorigenic (MCF-7, MDA-MB-231) and non-tumorigenic (MCF-10A) cell lines with compounds **2**, **4**, and **5** indicate that these substances are able to affect the ζ-potential of the cells. The anti-proliferative ability of compound **4** against the MCF-7 and MDA-MB-231 cell lines and its physico-chemical properties identify it as a lead compound.

## Data Availability

The data are stored by the authors and, if there is a valid reason, they can be sent via personal correspondence.

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
