# Peer review of "2-Alkyl-Substituted-4-Amino-Thieno[2,3-d]Pyrimidines: Anti-Proliferative Properties to In Vitro Breast Cancer Models"

_molecules, 2023, doi:10.3390/molecules28176347_

Round 1

Reviewer 1 Report

This MS reported synthesis of 2-alkyl-4-amino-thieno[2,3-d]pyrimidines, and screening for their cyto- and phototoxicity against BALB 3T3 cells, from which, lead compounds had been found to be potent against breast cancer cell lines. Cell cycle inhibition was assayed, and some druggable properties were virtually assessed.

1. In the introduction part, breast cancer therapy is lack of brief description, authors said “Despite the results achieved, no medicaments effective enough for the treatment of breast cancer have been developed yet”, which seems to be inconsistent with clinical used drugs for treatment of breast cancer.   

2. Drug design is based on the bioisosteres of heterocycles, but other substituents are un- rationally constructed. Substituent groups in of Gefitinib has not been fully referenced. The newly synthesized thieno[2,3-d]pyrimidines are too few (only have 4 compounds) to discuss and study the structure-activity relationships.

3. Figure 1 is too simple and should be deleted.

4. in the antiproliferation effect assays, there are no positive drug to be used as comparison.

So, I suggest this MS revised and submitted again.

Author Response

Thank you for your critical reading of our MS. By responding to your comments and questions, the quality of our research will be greatly improved.

This MS reported synthesis of 2-alkyl-4-amino-thieno[2,3-d]pyrimidines, and screening for their cyto- and phototoxicity against BALB 3T3 cells, from which, lead compounds had been found to be potent against breast cancer cell lines. Cell cycle inhibition was assayed, and some druggable properties were virtually assessed.

  1. In the introduction part, breast cancer therapy is lack of brief description, authors said “Despite the results achieved, no medicaments effective enough for the treatment of breast cancer have been developed yet”, which seems to be inconsistent with clinical used drugs for treatment of breast cancer.

The introduction is revised. A brief description of breast cancer therapy is also represented.

  1. Drug design is based on the bioisosteres of heterocycles, but other substituents are un- rationally constructed. Substituent groups in of Gefitinibhas not been fully referenced. The newly synthesized thieno[2,3-d]pyrimidines are too few (only have 4 compounds) to discuss and study the structure-activity relationships.

The choice of the substituents was based on the fact that a pharmacophore is the ensemble of steric and electronic features that is necessary to ensure necessary bioactivity. The choice of the substituents in the thienopyrimidine core is justified from this point of view as well as in relation to the structure of some of the already developed targeted breast cancer inhibitors. The motivation of the substituents is presented in the final paragraph of the Introduction.

We believe that the determinations of the physico-chemical properties of the compounds and the anti-proliferative results obtained confirm the choice made.

  1. Figure 1 is too simple and should be deleted.

Figure 1 was improved. We included some anti-breast cancer agents for showing the modification of basic structures.

  1. in the antiproliferation effect assays, there are no positive drug to be used as comparison.

As a positive control in antiproloferative assay we used cisplatin (CisPt), data are added in the text.

Reviewer 2 Report

Change Line 15-17 to : Thienopyrimidines are structural analogs of quinazolines and  the creation of new 2-alkyl derivatives of ethyl 4-aminothienopyrimidine-6-carboxylates for the study of their anti-proliferative properties is of a great pharmacological interest. 

Line delete the before WHO

Line 54 Change investigationя  investigations

Line 56 Change to: aiming for the generation...

Line 57 Change to: ...a number of thienopyrimidines  were synthesized as bioisosteres...

Line 79 leave space after that 2-(benz...

Delete sentence in lines 183 and 184 as it is repeated below in line 187

Line 201 what is SI  ???

Line 249 add full stop after diagnostic

Line 254 it is Figure 6

Line 255 It is Figure 6A

Line 387  give full name of acronym PAINS

Line 417 Insert space after products

Line 419 Change propanonitril to propanonitrile

In the materials and methods Section what does the character delta indicates in the IR spectra???

Also compounds are recrystallised from solvents. Change cases where describing recrystallization with a solvent!!

Line 567 The Zeta (ξ) potential... the character ξ (ksi) is wrong. It should be zeta (ζ )

In Supplementary Information the structure of the compounds should be inserted in the NMR spectra as shown in the IR spectra.  

Good proof-reading needed before re-submission.

Author Response

Thank you for your critical reading of our joint MS submitted to the journal Molecules.

Change Line 15-17 to: Thienopyrimidines are structural analogs of quinazolines and the creation of new 2-alkyl derivatives of ethyl 4-aminothienopyrimidine-6-carboxylates for the study of their anti-proliferative properties is of a great pharmacological interest.

It is changed

Line delete the before WHO

It is deleted

Line 54 Change investigationя  investigations

It is corrected

Line 56 Change to: aiming for the generation...

It is changed

Line 57 Change to: ...a number of thienopyrimidines  were synthesized as bioisosteres...

It is changed

Line 79 leave space after that 2-(benz...

It is added

Delete sentence in lines 183 and 184 as it is repeated below in line 187

It is deleted

Line 201 what is SI  ???

SI is selectivity index

Line 249 add full stop after diagnostic

It is added.

Line 254 it is Figure 6

It is corrected

Line 255 It is Figure 6A

It is corrected

Line 387  give full name of acronym PAINS

PAINS -- pan-assay interference compounds

Line 417 Insert space after products

It is inserted

Line 419 Change propanonitril to propanonitrile

It is corrected

In the materials and methods Section what does the character delta indicates in the IR spectra???

We used delta character to indicate bending vibrations in the IR spectrum.

Also compounds are recrystallised from solvents. Change cases where describing recrystallization with a solvent!!

It is done

Line 567 The Zeta (ξ) potential... the character ξ (ksi) is wrong. It should be zeta (ζ )

It is corrected

In Supplementary Information the structure of the compounds should be inserted in the NMR spectra as shown in the IR spectra. 

It is done

Author Response

Thank you your critical reading of our MS

  1. Typological error in Line 54

It is deleted

  1. In line 206, the IC50 value in molar form should be included

It is included

  1. For figure 3, the graph of reference compound should be included.

It is included. In our study, as a positive control for phototoxicity acridine orange (AO) was used. It is well known that AO possesses extremely high phototoxicity.

  1. Reference compound should be included for the anti-proliferation assay

It is included. As a positive control in the experiments for the anti-proliferative activity of the new compounds cisplatin (CisPt). The calculated selective index in the positive control (Cisplatin) is SI = 25.

  1. The concentration of the compounds used for the cell cycle study should be provided

The cells were treated with IC50.

  1. On Table 1, Table 2, Table 3 and Table 4, the decimal places should be unified

It is unified

  1. In line 230, the percentage of the untreated MCF-7 cells in the G1 phase shown in the

paragraph is different from what was listed on Table 4

It is corrected

  1. In Figure 6 it is hard to read the value for each column

It is corrected

  1. In line 273, compound 3 should be excluded, it didn`t show a lower anti proliferative activity against MCF-10A in comparison to the manifested cytotoxicity against MDA-MB-231 cell line.

It is corrected

  1. In line 307, space should be added for Figure6

It is corrected

  1. In line 590, the duplicated “IC50” should be deleted.

It is deleted

Round 2

Reviewer 3 Report

I appreciate the effort the authors have put into addressing the previous suggestions and concerns raised during the review process. The revisions have strengthened the paper and clarified several points that were previously ambiguous. However, there’s still one minor issue on line 284, it should be “compound 2,4 and 5”. After the revision, I think the paper is suitable for publication in Molecules.

Author Response

Dear Reviewer,

thank you for your thorough reading of our MS. The correction is made.